# Effect of Line Energy Conditions on Mechanical and Fatigue Properties of Ti6Al4V Fabricated by Electron Beam Additive Manufacturing

Youngsin Choi [1,2], Hwi-Jun Kim [1], Gun-Hee Kim [3], Chang-Woo Lee [1] and Dong-Geun Lee [4,*]

1    Korea Institute of Industrial Technology (KITECH), Incheon 21999, Korea; emailwnth@kitech.re.kr (Y.C.); khj@kitech.re.kr (H.-J.K.); cwlee@kitech.re.kr (C.-W.L.)
2    Department of Materials Engineering, Hanyang University, Ansan 15588, Korea
3    Gangwon Regional Division, Korea Institute of Industrial Technology (KITECH), Gangreung 25440, Korea; venkey@kitech.re.kr
4    Department of Materials Science and Metallurgical Engineering, Sunchon National University, Suncheon 57922, Korea
*    Correspondence: leechodg@scnu.ac.kr; Tel.: +82-61-750-3555

**Abstract:** Additive manufacturing has many process variables and requires additive process optimization. Line energy and scan speed are the main process variables. The objective of this work aims to investigate the effect of changes in line energy and scan speed among the process variables on the mechanical and fatigue properties of the Ti6Al4V specimens fabricated by electron beam additive manufacturing method. The size of the pore inside the specimen was 40~60 μm with the exception of the condition of 0.2 kJ/m, and the specimen with poor fusion of more than 100 μm and gas pore was found to have lower room temperature tensile and fatigue properties compared to the optimal process conditions. As line energy increased, strain hardening occurred, and yield strength and tensile strength increased. The $E_L$:0.3 kJ/m and 800 mm/s condition is a process condition that shows no defects such as unmelted powder and poor fusion, and it represents the best fatigue strength of 400 MPa. The fatigue strength of the specimen performed with hot isostatic pressing after additive manufacturing was measured at 550 MPa, an increase of 150 MPa, which resulted in high fatigue strength enhancement. The crack initiation site and propagation behavior were analyzed by observing the fatigue fracture section of the specimen according to the line energy.

**Keywords:** electron beam additive manufacturing; line energy; Ti6Al4V; process parameters; fatigue strength



## 1. Introduction

Additive manufacturing (hereinafter AM) technology using powder has been largely developed and has been used in various fields, including polymers, ceramics, stainless steel, Ti alloy, Co-Cr alloy, and Ni alloys for aircraft in recent years [1–8]. Polymer materials can be fabricated to variable shape easily but have low strength compared with additive manufacturing using metal powder. Ceramics have extraordinary hardness and moderate strength, but are difficult to fabricate some parts. Metals, in particular Ti6Al4V alloys, represent excellent properties such as ductility, strength, heat resistance, and corrosion resistance, and are therefore excellent materials that are widely accepted in the fields of energy and biomaterials.

An additive manufacturing process changes the mechanical and physical properties of fabricated materials according to different process variables. Thus, the optimization of the process variables is an important process. The powder bed fusion (PBF) method that builds specimens by radiating heat after spraying the powder onto a powder bed is a typical way of the selective laser melting (SLM) and electron beam melting (EBM) methods [9–14], and this study used the EBM method to conduct the additive manufacturing process.

Because the EBM method uses electron beam as a source of heat, it can radiate a wider area than the SLM method, which uses laser beams and causes rapid cooling [10]. The Ti6Al4V alloy manufactured by the EBM method causes rapid cooling at temperatures above the β transformation point and that produces martensite, which is the main cause of high strength, hardness, and low elongation of AM-printed parts [15,16]. Although EBM Ti6Al4V alloys have high strength values due to martensite, it represents lower fatigue properties than SLM.

In general, the specimens fabricated by the EBM method will have internal pore defects in which these pores are harmful to EBM AM-printed parts. The EBM process variables can be divided into seven main parameters: line energy, scan speed, line offset, focus offset, metal powder size, layer thickness, hatch spacing, and preheating. Although a high number of papers have been recently published on the evaluation of the properties of Ti6Al4V specimens fabricated by an AM process using powder [10,17–21], it is difficult to clearly determine the effect of the specific process variables in the seven different process parameters on the evolution of microstructure, quasi-static and dynamic mechanical properties, and formation of internal defects, etc. In the currently reported papers, it was confirmed that the tensile strength of the EBM-printed specimen using Ti6Al4V powders was 910 MPa and the high-cycle fatigue property (as specified in EN 6072 type 1) achieved $10^7$ cycles at 450 MPa [22].

In this study, the effect of changes in line energy among the EBM process variables on the mechanical and fatigue properties of the Ti6Al4V specimens were investigated.

The occurrence of defects (lack of fusion, gas pores, etc.) according to line energy variation could be confirmed through the analysis of microstructure and fatigue fracture surface of the specimen fabricated by the EBM additive manufacturing method. Therefore, a sound line energy condition was selected and in addition, the hot isostatic pressing (HIP) process was performed on the optimized process in order to evaluate the degree of increase in fatigue properties.

## 2. Materials and Methods

The property evaluation of Ti6Al4V rods fabricated by different conditions of line energy in the EBM process was performed. The range of process conditions was set through accumulated test know-how to satisfy both excellent quasi-static tensile properties and dynamic high cycle fatigue properties. The powder of the specimens used in this experiment was the Ti6Al4V powder manufactured by Arcam, GE Additive, Boston, Massachusetts, USA, and the powder sizes were d10 = 42 μm, d50 = 63 μm, and d90 = 93 μm. Table 1 shows the EBM fabrication condition applied to the specimens. The line energy ($E_L$) was varied in five different conditions, 0.2 kJ/m, 0.25 kJ/m, 0.3 kJ/m, 0.35 kJ/m, and 0.65 kJ/m, and the scan speed was controlled at 800 mm/s or 500 mm/s to fabricate rods. The microstructural properties, room temperature, tensile and mechanical properties were evaluated according to the five different line energy ($E_L$) conditions, and the $E_L$:0.3 kJ/m condition was selected as the optimal process condition by observing the fracture section according to fatigue failures through dynamic rotation bending fatigue test results. The hot isostatic pressure (HIP) postprocessing process was additionally performed on the selected optimal process conditions in order to compare the as-built specimens with the changes in mechanical and fatigue properties, and the HIP process was performed at 920 °C, 100 MPa, and 2 h.

**Table 1.** Process conditions used for the fabrication of EBM AM-printed Ti6Al4V samples.

|  | 0.2 kJ/m | 0.25 kJ/m | 0.3 kJ/m | 0.35 kJ/m | 0.65 kJ/m |
|---|---|---|---|---|---|
| Scan speed (mm/s) | 800 | 800 | 800 | 800 | 500 |
| Line energy (kJ/m) | 0.2 | 0.25 | 0.3 | 0.35 | 0.65 |
| Line offset (mm) | 0.1 | 0.1 | 0.1 | 0.1 | 0.1 |
| Layer thickness (μm) | 50 | 50 | 50 | 50 | 50 |
| Preheat (K) | 923 | 923 | 923 | 923 | 923 |

The specimens fabricated by the given process variables were cut by a wire cutting method, and the cross-section was etched after applying micro-polishing on the surface. Then, the observation and structure analysis were performed by using an optical microscope. The etching was implemented for 15 s~50 s by Kroll solution ($H_2O$ 100 mL, $HNO_3$ 5 mL, HF 2 mL), and the analysis of the microstructure properties was performed by using an OM Image Analyzer (LEICA ICC50HD, Wetzlar, Germany).

The cross-section of the cut specimen was ground and polished with a diamond polisher 6 μm → 1 μm, colloidal silica of 0.04 μm, and then the grain size and porosity changes through the HIP process were observed by an EBSD analyzer. EBSD measurements were analyzed with a field-emission scanning electron microscope (S-4300SE, HITACHI, Japan) and using TSL software(accessed on 22 April 2021) sets up a tolerance angle of 15° and confidence index value is higher than 0.1. In addition, the density measurement was performed using the Archimedes method (by physical law of buoyancy) according to the usual standard test on all specimens cut into Φ 15 mm and 10 mm sizes.

The Vickers hardness of the specimens was measured by the Vickers Hardness tester, Mitutoyo, and the mechanical properties were evaluated by pressing the conditions of 1 kgf, 10 s, and 11 points from the center to the side of the fabricated specimens were tested three times. The dynamic rotation bending fatigue tests were performed on more than 15 specimens by KDMT-240, Kyungdo Precision Co., Ltd.(Siheung-si, Gyeonggi-do, Korea) by using the round type specimens, as shown in Figure 1a–c and removing the specimens when the specimens were not fractured at 3000 rpm/min and $10^7$ cycles for obtaining high cycle fatigue strength. The fatigue fracture section was observed × 40 times and ×100 times by FE-SEM (JSM-7100F, JEOL, Japan).

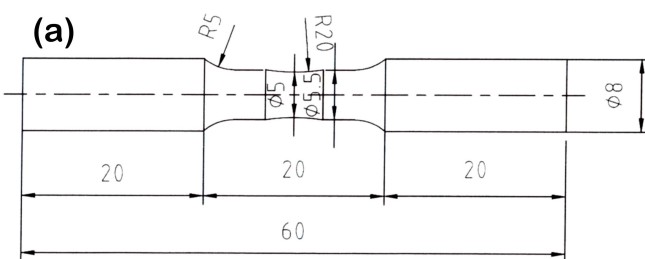

**Figure 1.** *Cont.*

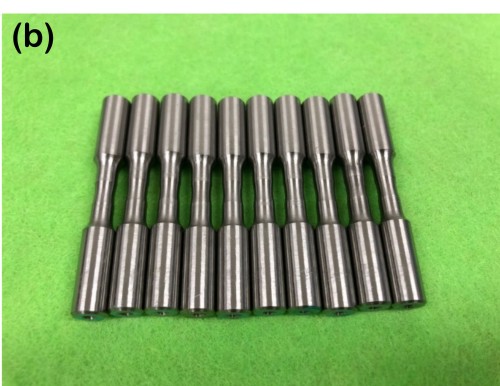

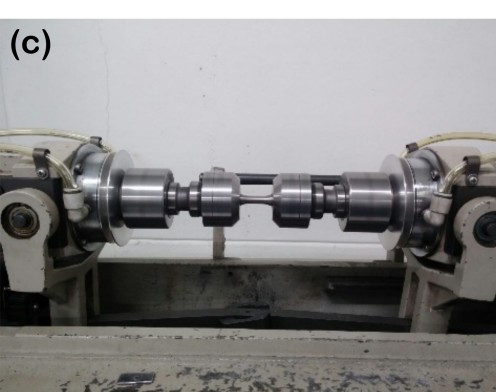

**Figure 1.** Illustration diagram and photos for dynamic rotation bending fatigue test; (**a**) schematic diagram of fatigue specimen, (**b**) machined fatigue specimens, and (**c**) dynamic rotation bending fatigue test system.

## 3. Results and Discussion

### 3.1. Microstructure Properties for Each Line Energy

Figure 2 shows the optical images of the sectional microstructure of Ti6Al4V fabricated rod specimens, which are fabricated by the EBM method by varying the five line energy conditions, 0.2 kJ/m, 0.25 kJ/m, 0.3 kJ/m, 0.35 kJ/m, and 0.65 kJ/m. As shown in Figure 2a, the specimen of 0.2 kJ/m was observed to have an unmelted defect with a scale more than 150 μm in a wide area near the edge of the fabricated rod. In the case of the specimen of 0.25 kJ/m with a slight increase in the internal heat input, the size of the internal defect was reduced compared to the specimen of $E_L$ 0.2 kJ/m in which the defect size was reduced to 45 μm (Figure 2b). The specimens of 0.3 kJ/m and 0.35 kJ/m showed no fusion defects like the specimen of 0.2 kJ/m and showed an average of less than 40μm of pore defects (Figure 2c,d). In the AM fabrication process with a scan speed of 800 mm/s, the process conditions of 0.3–0.35 kJ/m were shown to be suitable for the additive manufacturing by melting Ti6Al4V powder compared to the process conditions of 0.2–0.25 kJ/m. Thus, it has been verified that the fusion and pore defects inside the specimens can be reduced according to increases in the line energy. Figure 2e–h show the results of observing the microstructure of the specimens fabricated by the typical process conditions that conducted etching after applying a polishing process. Although decreases in the average pore defect size (150 μm → 45 μm → 40 μm) were verified according to increases in the line energy (0.2 kJ/m → 0.25 kJ/m → 0.3 kJ/m) through the microstructure observations, the specimen that has a high line energy of 0.65 kJ/m in the fabrication process with a scan speed of 500 mm/s, which is slower than 800 mm/s, showed the pore of about 60 μm, which is larger than the defects observed in the specimen of 0.3 kJ/m. In the case of the process condition of 0.65 kJ/m with reduced scan speed despite the positive effect of increased line energy, the defect in the specimen was greater than the defect under the conditions of 0.25–0.35 kJ/m. Thus, it shows that optimal line energy process conditions are needed to

reduce the pore defect size. Although it can be expected that more precise examinations of the Ti6Al4V powder applied with reduced scan speed can be expected, it was considered difficult to optimize the additive manufacturing condition by melting and solidifying the Ti6Al4V powder because of increasing pores due to the evaporation of Al element caused by large heat input [23]. Therefore, it was determined that the most stable microstructure conditions for the AM-printed specimens using the Ti6Al4V powder based on the EBM method were 0.3 kJ/m and 800 mm/s. The microstructure of the specimen processed by an additional process of the HIP processing for the specimens with the same fabrication condition of 0.3 kJ/m showed some grain coarsening, but there were no pores over 10 μm. Thus, it was possible to predict that it would have excellent high cycle fatigue properties.

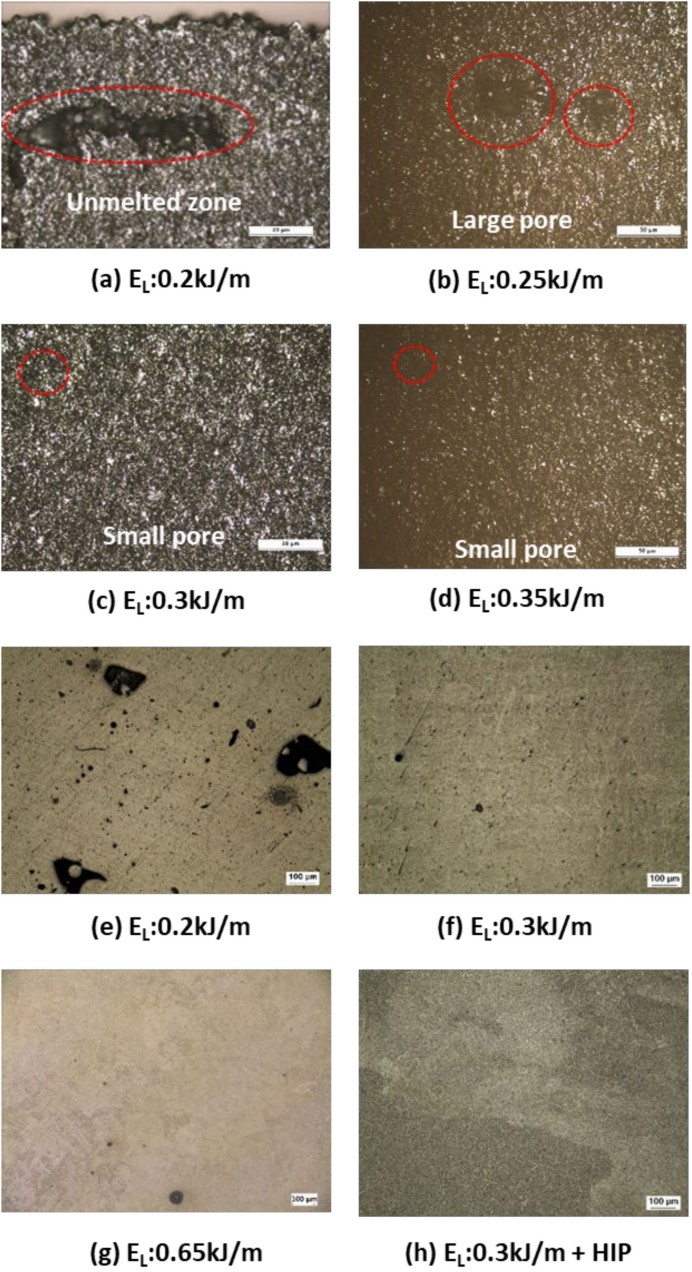

**Figure 2.** Optical micrograph images of the EBM AM-printed Ti6Al4V specimens before and after chemical etching; (**a**) $E_L$:0.2 kJ/m-before chemical etching, (**b**) $E_L$:0.25 kJ/m-before chemical etching, (**c**) $E_L$:0.3 kJ/m-before chemical etching, (**d**) $E_L$:0.35 kJ/m-before chemical etching, (**e**) $E_L$:0.2 kJ/m-after chemical etching, (**f**) 0.3 kJ/m-after chemical etching, (**g**) 0.65 kJ/m-after chemical etching, and (**h**) 0.3 kJ/m + HIP process-after chemical etching.

The EBSD data of the specimen before and after HIP treatment can be confirmed through Figure 3. Figure 3a–c showed the inverse pole figure EBSD map, phase map, and measured grain size distribution of the $E_L$:0.3 kJ/m condition among non-HIP specimens. Figure 3d–f are those of the $E_L$:0.3 kJ/m condition after HIP treatment. In Figure 3b, 0.009 fraction of beta phase was measured, and the total fraction of the alpha phase and the beta phase was 0.892. Accordingly, the pore fraction at the corresponding position was measured to be 0.108. In Figure 3c, there were almost no grains with over 15 μm size, and the average grain size of $E_L$:0.3 kJ/m was measured to be 5.17 μm. It can be seen that the fraction of the phase and pores after HIP treatment was significantly lowered when compared to that before HIP treatment, in Figure 3e. The total fraction of the alpha phase and beta phase was 0.992, and the measured pore fraction was 0.008. Through this, it was possible to confirm the reduction of porosity through HIP treatment. In Figure 3f, the average grain size was measured to be 8.57 μm, which is due to the grain growth by high treatment temperature in the HIP conditions (920 °C, 100 MPa). It was confirmed that after HIP, the number of grains with over 15 μm size increased when compared to the specimen before HIP treatment, and the fraction of beta phase did not differ significantly between before and after HIP treatment.

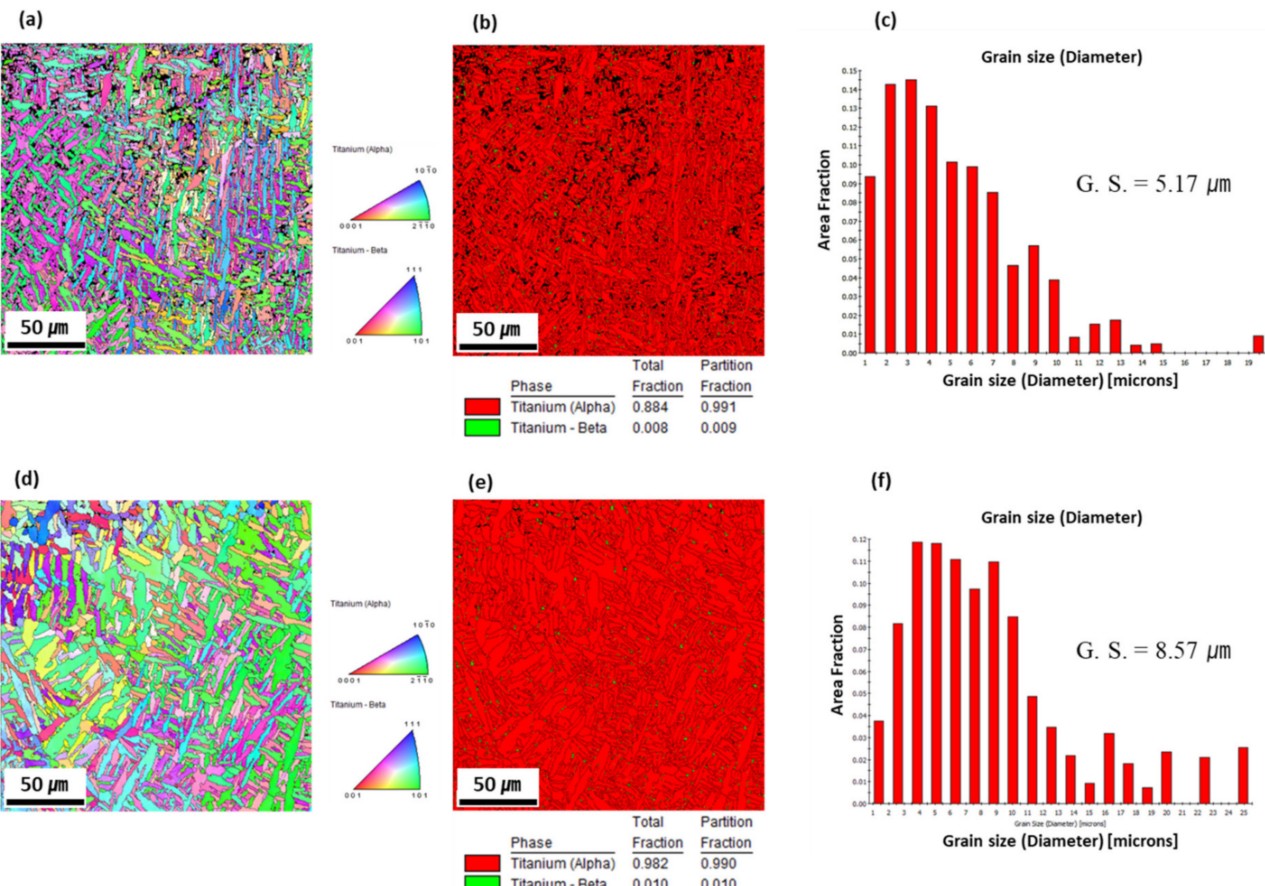

**Figure 3.** EBSD analysis before and after HIP treatment of the specimens: (**a**) inverse pole figure (IPF) EBSD maps of the non-HIPed specimen; $E_L$: 0.3 kJ/m (**b**) phase map of the non-HIPed specimen; $E_L$: 0.3 kJ/m (**c**) grain size of the non-HIPed specimen; $E_L$: 0.3 kJ/m (**d**) inverse pole figure (IPF) EBSD map of the specimen after HIP; $E_L$: 0.3 kJ/m (**e**) phase map of the specimen after HIP; $E_L$: 0.3 kJ/m (**f**) grain size of the specimen after HIP; $E_L$: 0.3 kJ/m.

### 3.2. Mechanical/Physical and Room Temperature Tensile Properties for Each Line Energy

The average Vickers hardness values of the specimens with a scan speed of 800 mm/s and fabricated with the process conditions of 0.2–0.35 kJ/m were measured as 357 Hv for the condition of 0.2 kJ/m, 365.5 Hv for the condition of 0.25 kJ/m, 354.6 Hv for the

condition of 0.3 kJ/m, and 352.4Hv for the condition of 0.35 kJ/m, and there was a tendency to slightly decrease in hardness according to increases in the line energy (Figure 4). If the amount of heat input for the same scan speed is highly determined, the cooling time will be relatively longer, which will affect the grain size. That is, as the amount of heat input increased, the grains were coarsened and that showed a decrease in hardness values. On the other hand, if the heat input supplied by the line energy is not sufficient (as the condition of 0.2 kJ/m), the internal unmelted zone may be widely distributed due to the insufficient melting and solidification of the fabricated powder, and the defects just below the surface may be present (as Figure 2a), which may contribute to the large decrease in hardness. This can be found in the description of the properties of the microstructure in Figure 2a. For the specimen of 0.65 kJ/m, it showed large increases in the heat input per hour due to the low scan speed. Thus, it represented a very small average hardness value, 346.3Hv, because of the greater coarsening in crystal grains. In addition, the HIP processed specimen was the specimen performing the pressurized heat treatment at 920 °C and the average hardness value of 340 Hv was measured. It represents a decrease in hardness due to the coarsening in crystal grains caused by the same mechanism as the preceding specimens.

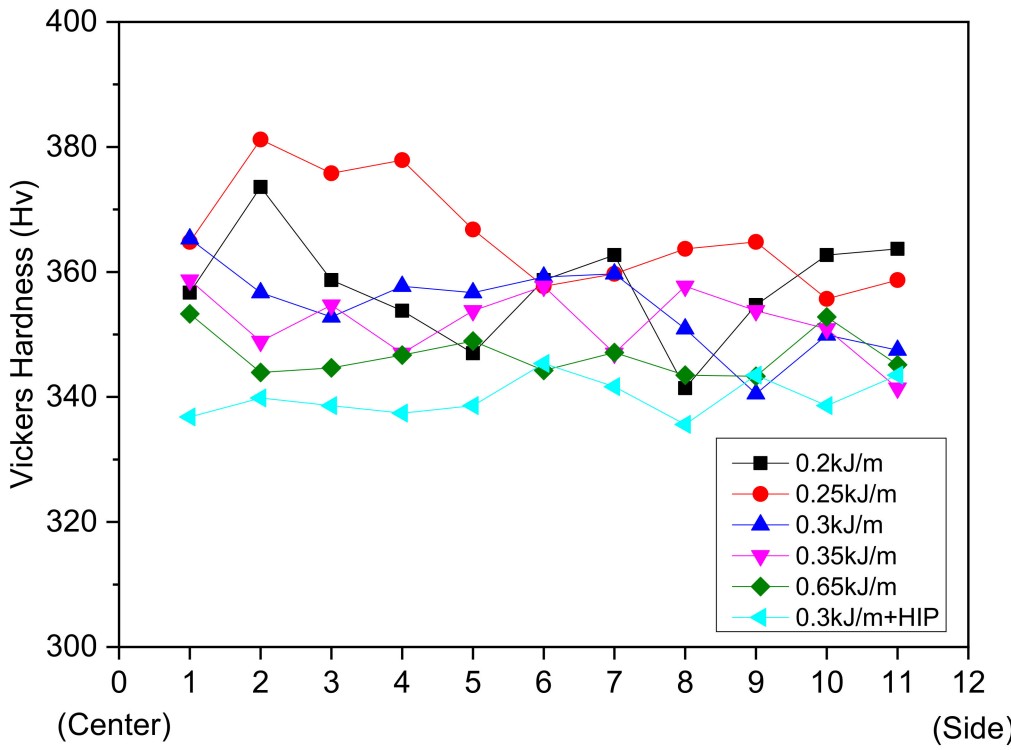

**Figure 4.** Vickers hardness of the EBM-printed Ti6Al4V specimens according to various line energy.

Figure 5 shows the room temperature tensile properties for each line energy condition. In case of 0.2 kJ/m, an accurate measurement was not made for the elongation and yield strength due to the incomplete internal condition as shown by the microstructure and hardness values. Thus, the elongation and yield strength of the specimen of 0.2 kJ/m is plotted by a slashed line in Figure 5 for reference as expected values based on the test data. The tensile strength of the comparison material, Ni-based superalloy 625, was 930 MPa [24], and the Ti6Al4V rod specimens fabricated by the EBM method showed superior tensile properties. The elongation of the fabricated specimens maintained at an excellent level of 15% or more, and this high elongation is one of the well-known properties of the AM-printed specimen using the EBM method [25].

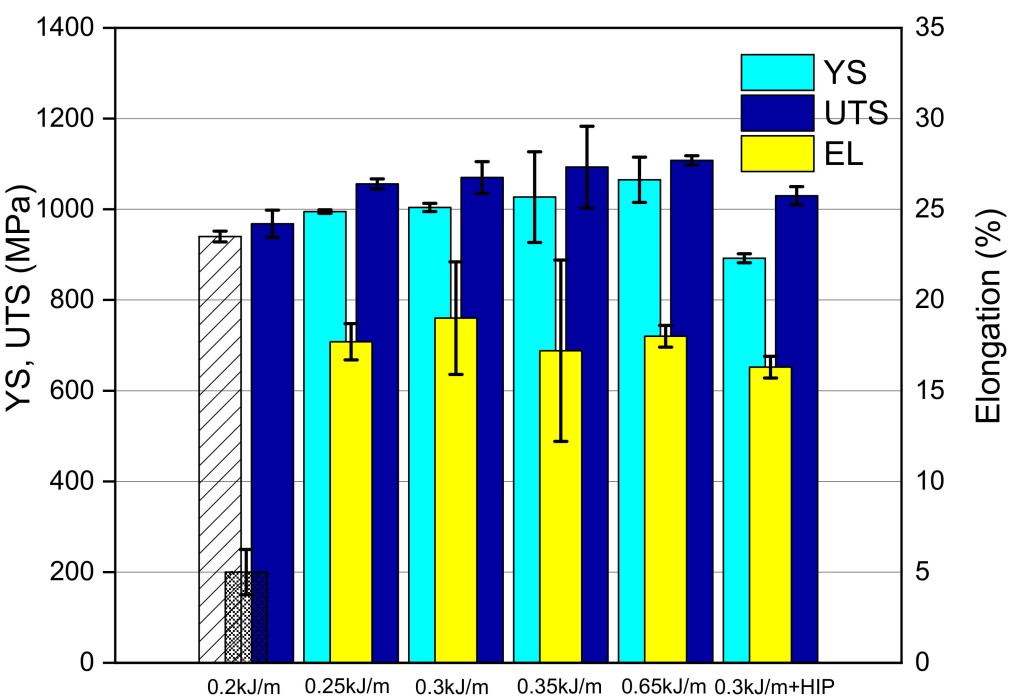

**Figure 5.** Tensile properties for all conditions of the EBM process.

In the conditions of $E_L$ = 0.25–0.35 kJ/m, a tendency to increase tensile strength and yield strength according to increases in the line energy was shown. This represents an inverse proportional result to the tendency of the Vickers hardness, and it can be explained by the strain hardening effect mechanism. As the line energy increases, the residual heat remaining in the previous layer causes an annealing effect and that causes strain hardening during a plastic deformation process under tensile loading [26]. Thus, the more the line energy, the higher the yield strength and tensile strength. In addition, as the strain hardening effect increases, the effect of hardness values on yield strength decreases. Typical materials show a proportional relationship between hardness and yield and tensile strength, but those with a high strain hardening effect or very low hardness and yield strength do not have a proportional relationship [27]. In contrast to the results of the microstructural analysis described in Section 3.1, the results of the room temperature tensile test may not be proportional to the effect of increase of line energy and grain coarsening. In addition, it is deemed that pores less than 100 μm do not significantly affect the results of the room temperature tensile test.

In case of the fabrication process with a scan speed of 800 mm/s, the more line energy, the greater the error in reproducibility. Furthermore, this can be confirmed through the error bar in the graph of the room temperature tensile test. As the fabrication process was performed at a scan speed of 500 mm/s, despite the line energy 0.65 kJ/m, the reproducibility and room temperature tensile properties (yield strength, tensile strength, and elongation) were good enough to titanium wrought alloys. In addition, despite the large defect size compared to other process conditions, it was confirmed that the quasi-static properties are stable through the tendency to maintain the room temperature tensile properties. However, it was confirmed that the conditions of 800 mm/s and $E_L$: 0.3 kJ/m are the more optimum conditions in the minimization of defects, high room temperature tensile properties, and reproducibility of fabrication than that of the conditions of 500 mm/s and $E_L$: 0.65 kJ/m because of decrease in hardness values caused by the productivity and heat input of the process. In the case of applying the specimen treated by the HIP process to the specimen of the condition of $E_L$: 0.3 kJ/m, which was selected as the optimum condition, it shows the coarsening in crystal grains and that causes a decrease in yield strength because it is processed by a heat treatment process with high temperature more

than 900 °C. On comparing it with the yield strength of the as-built specimen condition of $E_L$: 0.3 kJ/m, it was verified that about 13% of strength was decreased.

The results of density measurement by the Archimedes method are given in Table 2. The average density of Ti6Al4V was 4.43 g/cm$^3$, and the pore volume was measured based on the corresponding value. The density of $E_L$: 0.2 kJ/m conditions was the lowest as 4.219 g/cm$^3$, and the density was measured as 4.275 g/cm$^3$ under $E_L$: 0.3 kJ/m conditions. From the data, it was confirmed that the densities of the as-fabricated specimens without HIP treatment increased in the order of 0.2 kJ/m → 0.25 kJ/m → 0.35 kJ/m → 0.3 kJ/m in line energy. This indicated the superiority of the $E_L$: 0.3 kJ/m condition in EBM lamination, and the porosity under that condition was 3.5% compared to the average density of the Ti6Al4V specimen.

**Table 2.** Density and pore fraction of the EBM-printed Ti6Al4V samples fabricated according to various line energy.

| Line Energy Conditions | Weight [g] | Water + Specimen [g] | Temperature [K] | Measured Density [g/cm$^3$] | Pore Volume Fraction [%] |
|---|---|---|---|---|---|
| 0.2 kJ/m | 0.8514 | 0.6498 | 290 | 4.219 | 4.77 |
| 0.25 kJ/m | 1.1018 | 0.8411 | 290 | 4.223 | 4.68 |
| 0.3 kJ/m | 0.9574 | 0.7337 | 290 | 4.275 | 3.50 |
| 0.35 kJ/m | 0.9820 | 0.7522 | 290 | 4.268 | 3.67 |
| 0.65 kJ/m | 0.9909 | 0.7579 | 290 | 4.248 | 4.10 |
| 0.3 kJ/m + HIP | 0.9983 | 0.7780 | 290 | 4.526 | –2.16 |

The specimens of $E_L$: 0.2 kJ/m–0.25 kJ/m conditions were found to have unmelted zone defect and the pore volume fraction was measured to be over 4.68%. Therefore, the lamination condition is judged to be an incomplete condition for the melting of the powder, and it proves that line energy needs to be improved for the lamination of sound specimens. Therefore, the lamination condition is considered to be an incomplete condition for the melting of the powder, and it means that the line energy needs to be increased for the lamination of a sound specimen. The pore distribution before HIP treatment can be confirmed in the phase map in Figure 3. $E_L$: 0.3 kJ/m +HIP specimen had a density of 4.526 g/cm$^3$, which was greater than the average density of Ti6Al4V. When the analysis was conducted in terms of understanding the tendency of density change under each condition, it was confirmed that the porosity of the HIP-treated specimen was significantly decreased, which was consistent with the data in Figure 3.

### 3.3. High Cycle Fatigue Properties for Each Line Energy

Figure 6 shows the results of the high cycle dynamic rotation bending test at a stress level more than 400 MPa. The specimen of $E_L$: 0.3 kJ/m achieved the number of rotations more than $10^7$ without being destroyed under the stress of 400 MPa. This value is less than one-half of the room temperature tensile value and can be verified to be excellent compared with Ni-based superalloy 625 (tensile strength: 930 MPa, fatigue strength: 380 MPa) [24]. The specimen of $E_L$: 0.65 kJ/m was destroyed at a stress of 450 MPa with the number of rotations recorded as $2 \times 10^5$, and that was a similar value compared to the specimen of 0.3 kJ/m under the same stress. The additional tests for the specimen of 0.2k J/m were stopped due to several internal defects observed in the fracture section. The specimen of 0.25 kJ/m showed fatigue failures at the number of rotations less than $10^5$ in different loads under 400 MPa. Except for the conditions of the scan speed of 800 mm/s and line energy of 0.3 kJ/m, the other conditions did not achieve the number of rotations, $10^7$, that did not result in failure under a load of 400 MPa or more, and the results of these dynamic fatigue failure were found to be different from the results of the quasi-static tensile test. The fatigue strength of the specimen processed by the condition of 0.3 kJ/m + HIP was measured

as 550 MPa that shows an increase of 150 MPa for the specimen of 0.3 kJ/m recorded as 400 MPa. This shows that the fatigue strength improvement through the HIP treatment is excellent at 37.5% when compared with the increase in the fatigue strength by 5% [28] as the increase in the fatigue strength through a post heat treatment is 20 MPa. Therefore, the HIP processing process is more effective than the post heat treatment process [28] in order to significantly improve the high cycle fatigue properties of AM-printed products fabricated by the EBM method.

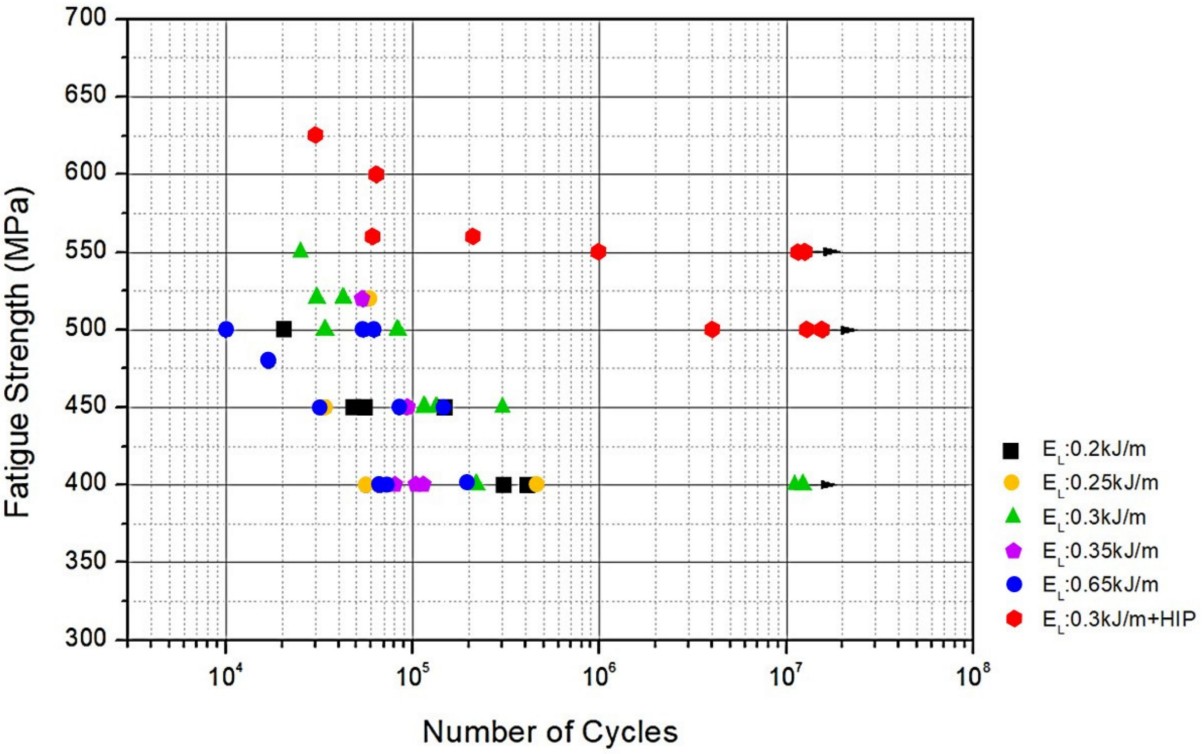

**Figure 6.** HCF-fatigue data from the test of the EBM AM-printed Ti6Al4V with the HIP process.

Figure 7 represents the images of the fracture sections of the specimens of 0.2k J/m, 0.3 kJ/m, 0.65 kJ/m, and 0.3 kJ/m + HIP observed by using SEM. As shown in Figure 7a, lots of unmelted powder can be observed and no specific crack initiation sites could be found. It is possible to verify that the specimen has low room temperature tensile properties and fatigue strength values due to the large number of unmelted powder areas distributed inside and on the surface of the specimen. As shown in Figure 7b, the specimen of 0.3 kJ/m was cracked in two areas marked with squares and showed a typical fatigue fracture shape. In addition, about 40 μm of gas pores were observed as in the microstructure described earlier. The crack initiated in the left side was caused by internal inclusions, and the crack initiation in the right side was caused by the pores located below the surface. The specimens of 0.2 kJ/m and 0.3 kJ/m showed that the fatigue strength can be reduced when unmelted powder is presented through the fatigue fracture section. Therefore, it is necessary to pay attention to the hatch spacing and overlap of melting section during its lapping process in order to avoid unmelted powder. In addition, although the pores located just below the surface are the same size, it can be observed that it acts as a crack initiation site as a priority over the 40 μm of pores located inside the specimen. In the specimen of 0.65 kJ/m presented in Figure 7c, a number of about 53 μm pores were distributed, and in addition to the pores, micro-cracks could be observed. Therefore, it is considered unstable as compared with the process condition of the specimen of 0.3 kJ/m. The specimen of 0.3 kJ/m + HIP (Figure 7d) showed the initiation of cracks due to the fusion defect located below the surface, but there were no internal pores unlike the as-fabricated specimen of

0.3 kJ/m and the striations were prominent. The HIP process carried out in this experiment was effective in removing internal pores from the specimen of 0.3 kJ/m and resulted in a corresponding increase in fatigue strength.

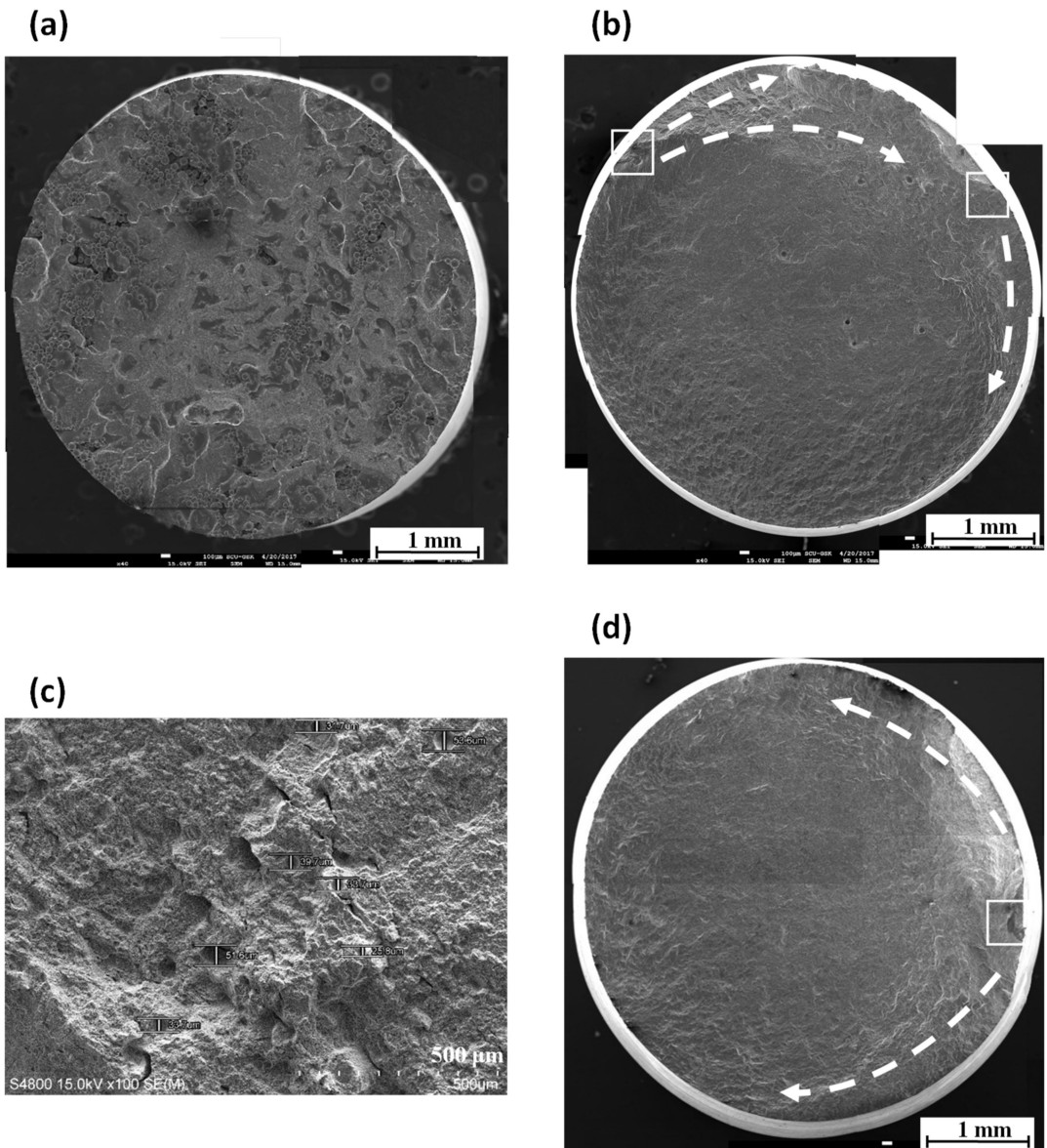

**Figure 7.** SEM of fracture surfaces of the EBM AM-printed Ti6Al4V with different fabrication conditions; (**a**) EL:0.2 kJ/m, (**b**) EL:0.3 kJ/m, (**c**) EL:0.65 kJ/m, and (**d**) EL:0.3 kJ/m + HIP process.

## 4. Conclusions

1. AM-printed bulk rods were fabricated by the EBM method through a Ti6Al4V powder fabrication process under various line energy process conditions. After analyzing the microstructure for the specimens fabricated through five process conditions, it was confirmed that as the line energy increases, the fusion and pore defects inside the specimens can be reduced. In particular, in the case of the specimen of 0.3 kJ/m, a small amount of pores of about 40 µm was found in microstructure and fatigue fracture sections and no other defects such as unmelted powder or poor fusion were found.

2. The condition for the highest Vickers hardness value was $E_L$: 0.2 kJ/m, and the lowest hardness value was $E_L$: 0.65 kJ/m, and as the line energy increased, the Vickers

hardness value decreased. This is caused by the coarsening of crystal grains due to an increase in the amount of heat input. On the other hand, as the line energy increased, the yield strength increased because a strain hardening effect is generated by the annealing effect caused by the residual heat in the previous layer during the fabrication process.

3.  According to the results of the dynamic rotation bending fatigue test of the specimen of Ti6Al4V, the specimen of 0.3 kJ/m represented a fatigue strength value of 400 MPa and achieved a higher fatigue strength than other processes. Considering internal defects, hardness, room temperature tensile and fatigue properties, the proper fabrication process conditions for the EBM AM process of the Ti6Al4V powder were the line energy of 0.3 kJ/m and scan speed of 800 mm/s. Then, the fatigue strength of the specimen of 0.3 kJ/m + HIP, which was processed by HIP, was improved effectively to 550MPa, increased by 150MPa from the as-fabricated specimen of 0.3 kJ/m.

**Author Contributions:** Conceptualization, Y.C. and D.-G.L.; methodology, Y.C., H.-J.K. and G.-H.K.; validation, H.-J.K., G.-H.K., C.-W.L. and D.-G.L.; formal analysis, Y.C. and D.-G.L.; investigation, Y.C. and D.-G.L.; writing—original draft preparation, Y.C. and D.-G.L.; writing—review and editing, D.-G.L.; funding acquisition, D.-G.L., H.-J.K. and C.-W.L. All authors have read and agreed to the published version of the manuscript.

**Funding:** This work was supported by the Korean government MOTIE (the Ministry of Trade, Industry and Energy), the Korea Evaluation Institute of Industrial Technology (KEIT) (No. 10053101) and the Korea Institute for Advancement of Technology (KIAT) (No. P0002019).

**Data Availability Statement:** The data presented in this study are available on request from the corresponding author.

**Conflicts of Interest:** The authors declare no conflict of interest.

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
