# Peer review of "Effect of Line Energy Conditions on Mechanical and Fatigue Properties of Ti6Al4V Fabricated by Electron Beam Additive Manufacturing"

_metals, doi:10.3390/met11060878_

Round 1

Reviewer 1 Report

The work of Choi et al. studies two parameters of the EBAM 3D manufacturing and their effects in the typical commercial Ti6Al4V alloy for biomedical and aeronautical applications.

In my opinion, although in general the information was well organized, and the methodology proposed and carried out is appropriated, this work need to be clearly improved to be suitable for publication in metals journal. The criticisms are following exposed:

  1. The written of this manuscript must be clearly improved. As example:

1a) In the abstract, the word “process” is written three times in the same sentence.

1b) Please, remove the acronyms and symbols in the abstract.

1c) Line 33: “Additive manufacturing (hereinafter 3D printing)”. However, the concept “additive manufacturing” is more “extensive” than “3D printing”, because not all AD techniques are 3d printing. In addition, during the work, another times authors refers as additive manufacturing when they inform that from this point, they are going to name as 3D printing. Please, correct this aspect and modify it adequately.

1d) Line 38: “to fabricating” it is not correct. “fabricating” or “to fabricate”.

1d) Line 37: Modify “Ceramics has extraordinary” by “Ceramics HAVE extraordinary”.

1e) Line 58: Modify “Although a number of papers” by “Although a HIGH number…”

1f) Line 122: replace “incomplete fusion” by “unmelted”.

1g) Line 149: replace micro-structure for microstructure.

1h) Line 155: What does mean IPF? Please, introduce it in the manuscript.

Thus, please, check, modify and improve the written of the entire manuscript.

  1. In the abstract section, the line 25 and 26 are confused because it seems that the improved specimen is only fabricated by HIP and not by EBAM plus HIP. Please, correct and improve.

3) Replace Ti-6Al-4V by Ti6Al4V throughout the manuscript. Usually, in metallurgy and materials science, the hyphen is used for amorphous alloys and not crystalline alloys.

4) In the experimental section, mor data related to the processing is necessary, such as why the EL and scan speed conditions, selected, why the HIP conditions, why authors are used only the HIP for a single specimen, number of indentations for Vicker`s Hardness, numbers of tests for mechanical testing and fatigue tests, etc. Also, the standard used for Archimedes method and the procedure applied. In addition, for Kroll Solution, the immersion time, etc.

5) The quality of the polished SEM images must be improved without polishing lines.

6) Line 151: How have the authors measured the coarsening of the particles from figures 2? It is impossible to measure this aspect from SEM images because some particles can be formed by different crystal domains. It would be necessary DRX, electron diffraction, etc. It is better to say some particle coarsening instead of coarsening crystals.

7) Lines156-157. Similar comment to the comment 6. How have authors determined the fraction of alpha and beta phases? It would be necessary the study of XRD. In my opinion, it is mandatory the presence of XRD for all specimens, to determine the phases, possible segregation of phases, crystalline domain size, etc.

8) Related to mechanical properties, concretely the Vickers tests, lines 185-189, What is the difference in hardness attributable to? Please explain. In my opinion, it is necessary to show a HV vs gran size graphic to determine the tendency.

9) What is the difference in center and side hardness attributable to? Please, explain.

10) The standard deviation has to be showed in the figure 4. Thus, it would be possible observe how there is no significance difference between 340 and 346 Hv.

11) The extensive paragraph evolved lines 207-233 it is really difficult to understand. Please, modify and improve it.

Author Response

- The revised manuscript can be improved, compared to its original version. We appreciated that the manuscript has been improved by the Reviewers’ comments.

1) The written of this manuscript must be clearly improved. As example:

- According to the Reviewer’s comment as below, we revised the manuscript.

1a) In the abstract, the word “process” is written three times in the same sentence.

1b) Please, remove the acronyms and symbols in the abstract.

1c) Line 33: “Additive manufacturing (hereinafter 3D printing)”. However, the concept “additive manufacturing” is more “extensive” than “3D printing”, because not all AD techniques are 3d printing. In addition, during the work, another times authors refers as additive manufacturing when they inform that from this point, they are going to name as 3D printing. Please, correct this aspect and modify it adequately.

1d) Line 38: “to fabricating” it is not correct. “fabricating” or “to fabricate”.

1d) Line 37: Modify “Ceramics has extraordinary” by “Ceramics HAVE extraordinary”.

1e) Line 58: Modify “Although a number of papers” by “Although a HIGH number…”

1f) Line 122: replace “incomplete fusion” by “unmelted”.

1g) Line 149: replace micro-structure for microstructure.

1h) Line 155: What does mean IPF? Please, introduce it in the manuscript.

Thus, please, check, modify and improve the written of the entire manuscript.

2) In the abstract section, the line 25 and 26 are confused because it seems that the improved specimen is only fabricated by HIP and not by EBAM plus HIP. Please, correct and improve.

- According to the Reviewer’s comment, we revised the abstract.

3) Replace Ti-6Al-4V by Ti6Al4V throughout the manuscript. Usually, in metallurgy and materials science, the hyphen is used for amorphous alloys and not crystalline alloys.

- According to the Reviewer’s comment, we replaced Ti-6Al-4V by Ti6Al4V throughout the manuscript.

4) In the experimental section, mor data related to the processing is necessary, such as why the EL and scan speed conditions, selected, why the HIP conditions, why authors are used only the HIP for a single specimen, number of indentations for Vicker`s Hardness, numbers of tests for mechanical testing and fatigue tests, etc. Also, the standard used for Archimedes method and the procedure applied. In addition, for Kroll Solution, the immersion time, etc.

- According to the Reviewer’s comment, we revised the experimental section of the manuscript.

HIP conditions were selected as the well-known HIP condition for Ti6Al4V alloy by considering the properties of titanium alloy and the previous researches to reduce the porosity.

For this study, the best line energy condition was selected, and HIP process was performed to further improve the HCF property of the selected condition specimens.

We revised the test conditions of etching, hardness and mechanical tests, etc.

When the line energy is too high and the scan speed is too late, input heat amount becomes too high and evaporation phenomenon can be occurred. As a result, large and many void defects inside matrix can be formed and affect the negative mechanical properties. The range of process conditions was set through the accumulated test know-how. There are optimum conditions to satisfy both excellent quasi-static tensile properties and dynamic fatigue properties. This study was carried out in the appropriate condition ranges especially for high cycle fatigue properties.

Thanks for your comments.                                                                  

5) The quality of the polished SEM images must be improved without polishing lines.

- We are sorry that it was difficult to completely remove the polishing line due to the pores inside matrix. Please understand this difficulty widely.

6) Line 151: How have the authors measured the coarsening of the particles from figures 2? It is impossible to measure this aspect from SEM images because some particles can be formed by different crystal domains. It would be necessary DRX, electron diffraction, etc. It is better to say some particle coarsening instead of coarsening crystals.

- We revised the wrong expressions. When the amount of heat input by HIP increased, the grains were coarsened and hardness values were decreased, as shown in Fig. 4.

Grain coarsening after HIP treatment was checked by grain size distribution and average grain size through EBSD analysis method, as shown in Fig. 3(c) and (f).

7) Lines156-157. Similar comment to the comment 6. How have authors determined the fraction of alpha and beta phases? It would be necessary the study of XRD. In my opinion, it is mandatory the presence of XRD for all specimens, to determine the phases, possible segregation of phases, crystalline domain size, etc.

- The fractions of alpha and beta phases were calculated and measured automatically by general EBSD analysis method, as shown in Fig. 3(b) and (e). Red color region means alpha phase region and green color region beta phase.

8) Related to mechanical properties, concretely the Vickers tests, lines 185-189, What is the difference in hardness attributable to? Please explain. In my opinion, it is necessary to show a HV vs gran size graphic to determine the tendency.

- There was a tendency to slightly decrease in hardness as according to increases in the line energy (Figure 4). If the amount of heat input for the same scan speed is highly determined, the cooling time will be relatively longer, which will affect the grain size. That is, as the amount of heat input increased, the grains were coarsened and that showed a decrease in hardness values. On the other hand, if the heat input supplied by the line energy is not sufficient (as the condition of 0.2 kJ/m), the internal unmelted zone may be widely distributed due to the insufficient melting and solidification of the fabricated powder, and the defects just below the surface may be present (as Fig. 2(a)), which may contribute to the decrease in hardness.

9) What is the difference in center and side hardness attributable to? Please, explain.

- Since ‘micro hardness’ is measured, errors may occur depending on the state of the microstructure at each location of the matrix. As line energy increased from 0.2 kJ/m to 0.65 kJ/m, macroscopical average hardness changes decreased. The 0.3 kJ/m+HIP specimen showed the lowest hardness because it was taken in high temperature HIP treatment (Fig. 4).

10) The standard deviation has to be showed in the figure 4. Thus, it would be possible observe how there is no significance difference between 340 and 346 Hv.

- Vickers hardness graph expressed the values measured by 11 points each specimen, and by obtaining the average of these values, the hardness characteristics for each line energy can be evaluated. 340 Hv and 346 Hv values showed the average hardness value.

Please understand widely that the error bars are not inserted to alleviate the complexity of the complicated graphs.

11) The extensive paragraph evolved lines 207-233 it is really difficult to understand. Please, modify and improve it.

- According to the Reviewer’s comment, we revised the manuscript. Thanks for your kind comment. 

Reviewer 2 Report

Line 57 - When you talk about EBM technology , and you mention process parameter, that influences it, you must talk about the  size of metals powder grains, as an important parameter. The paper does not include  this variable in the list of parameters, although Line 79 specifies the  size of 3 types of grains.  The authors must justify why they did not entered this parameter  in the considered list, and don't mention the used granules.

Line 65 - Must be specified the type of fatigue test used and the value of the angles. Must be also specified all the parameters of the testing machine used.

Line 81 - On what basis were the 5 mentioned energies lines chosen. Same question for the two scan speeds.

Line 89 - What criteria did you used to choose the HIP process parameters ?

Tab 1 - Scan speed cannot be considered a parameter if it has only 2 values ( 500mm/s only one time) and not those technologically justified.

Tab 1 - Layer thickness - if it has a constant value it cannot be considered a parameter so it has nothing to look for in the parameter table. So, the same is the heating temperature. 

Line 110 - How long is the angle per cycle?

Line 138 - It was necessary to try with 800 mm/s with the power of  0.65 KJ/m in order to make  correct comparisons.

Line 144 -  This is not about Ti powder. It is  Ti-6Al-4V powder.

Line 163 - It is a natural conclusion.  It is normal that after pressing at  temperature, the grains increase in size.  What is the importance of this conclusion?

Fig.4 - Why the Vickers hardness at 0.25 KJ/m is higher than the Vickers hardness at 0.20 KJ/m?  The fact that the law stated by you above: "... increasing the energy decreases hardness..."  is not respected, must be explained.

Line 258 - You compare densities obtained under different conditions. Remove the 0.65 KJ/m because it use a different speed.

Tab. 2 - Contains  a serious error, because there is not correlation between weight and density, although it should be.  The exceptions from tab. 2  must be correlated and explained otherwise  you violate the laws of  physics.

Lines 279 - 292 - The comments made do not have a statistical basis. In order to draw such conclusions, at least 9 specimens had to be  tried and processed in the same conditions, at each energy line. Repeat section 3. 3 under the conditions proposed above, based on  a reliable statistical analysis.

Line 297 - Explain what "...post heat treatment..." consisted of???

Fig. 6 - The data entered in the diagram are insufficient for statistical interpretation. This interpretation is mandatory if we take into account the strong random character of the formation, by local melting,  of the powders and their transformation into specimens. There is also a strong random character of the non-uniformities obtained between specimens that are not even analyzed. The above are supported even by the authors in the comments of the data presented in fig 7, without mentioning whether they are average of values or values obtained in a single measurement.

Conclusion - 1 -In the paper there not were : "...various scan speed...". It was a single 800mm/s. The ones of 500 mm/s is unconvincing because only one attempt was made with it. It would be normal that at a lower scanning speed (500 mm/s) and with a higher energy THAN OTHERS USED (0.65 kj/M) THESE SPECIMENS WOULD BE THE BEST MELTED, fact that does not result from he analysis of the presented data.

General observations: The paper requires a better definition of variable parameters, a better explanation of internal phenomena and a better correlations between parameters and values obtained by using statistical analysis of experimental data, analysis imposed by the random nature of the processes that take place.

Author Response

- The revised manuscript can be improved, compared to its original version. We appreciated that the manuscript has been improved by the Reviewers’ comments.

Line 57 - When you talk about EBM technology , and you mention process parameter, that influences it, you must talk about the  size of metals powder grains, as an important parameter. The paper does not include  this variable in the list of parameters, although Line 79 specifies the  size of 3 types of grains.  The authors must justify why they did not entered this parameter  in the considered list, and don't mention the used granules.

- According to the Reviewer’s comment, we inserted the metal grain size. We agree that the reviewer's opinion is valid, although it refers mainly to the parameters of the equipment process aspect. Thanks for your good comment.

Line 65 - Must be specified the type of fatigue test used and the value of the angles. Must be also specified all the parameters of the testing machine used.

- According to the Reviewer’s comment, we inserted the information. We can check out the type of the fatigue test in a paper published by Alexander K et al (Ref. 22).

Alexander K, Burghardt K, Thomas W, Bernd K, et al reported the HCF properties of Ti6Al4V alloy in the paper of “Mechanical Properties of Ti-6Al-4V Fabricated by Electron Beam Melting”. In this paper, the specimens for high-cycle fatigue test were 90 mm long and 2.5 mm thick with a minimum width of 7 mm as specified in EN 6072 type 1.

Line 81 - On what basis were the 5 mentioned energies lines chosen. Same question for the two scan speeds.

- When the line energy is too high and the scan speed is too late, input heat amount becomes too high and evaporation phenomenon can be occurred. As a result, large and many void defects inside matrix can be formed and affect the negative mechanical properties. The range of process conditions was set through the accumulated test know-how. There are optimum conditions to satisfy both excellent quasi-static tensile properties and dynamic fatigue properties. Scan speed is mainly 800 mm/s, and 0.65kJ/m+500mm/s is simply for additional analysis of the case of large heat input. This study was carried out in the appropriate condition ranges especially for high cycle fatigue properties.  

Thanks for your comments.

Line 89 - What criteria did you used to choose the HIP process parameters ?

- HIP conditions were selected as the well-known HIP condition for Ti6Al4V alloy by considering the properties of titanium alloy and the previous researches to reduce the porosity.

Tab 1 - Scan speed cannot be considered a parameter if it has only 2 values ( 500mm/s only one time) and not those technologically justified.

- We agree that the Reviewer’s comment. Scan speed was not considered as a main parameter. As can be seen from the title of this manuscript, the main parameter is line energy, which is to investigate its effect. Scan speed is mainly 800 mm/s, and only 0.65kJ/m+500mm/s is simply for additional analysis of the case of large heat input.

Tab 1 - Layer thickness - if it has a constant value it cannot be considered a parameter so it has nothing to look for in the parameter table. So, the same is the heating temperature. 

- According to the Reviewer’s comment, we revised the caption of Table 1. Table 1 expresses test process conditions, not process parameters. Thanks for your kind comment.

Line 110 - How long is the angle per cycle?

- Dynamic rotation bending test for high cycle fatigue, as shown in Fig. 1(a)~(c), is a method of testing by hanging weights with different loads using a standard round type specimen. Dynamic rotation bending tests were carried out at 3000 rpm/min under each load and stopped when the specimens were not fractured at over 107 cycles for obtaining high cycle fatigue strength.

Line 138 - It was necessary to try with 800 mm/s with the power of  0.65 KJ/m in order to make  correct comparisons.

- The main parameter is line energy. Scan speed is 800 mm/s, and 0.65kJ/m+500mm/s is simply for additional comparison analysis of the case of large heat input by raising line energy and lowering scan speed.

Line 144 -  This is not about Ti powder. It is  Ti-6Al-4V powder.

- According to the Reviewer’s comment, we revised the manuscript.

Line 163 - It is a natural conclusion.  It is normal that after pressing at  temperature, the grains increase in size.  What is the importance of this conclusion?

- From the EBSD analysis in Figure 3, it was possible to check the grain size, phase fraction and porosity before and after HIP treatment. Internal pores can be removed and the density improved through HIP treatment, and it is confirmed that HIP treatment is effective to improve the dynamic high-cycle fatigue strength. EBSD, phase map, and grain size distribution of the non-HIPed and the HIPed specimens (EL: 0.3 kJ/m) were investigated.

Fig.4 - Why the Vickers hardness at 0.25 KJ/m is higher than the Vickers hardness at 0.20 KJ/m?  The fact that the law stated by you above: "... increasing the energy decreases hardness..."  is not respected, must be explained.

- According to the Reviewer’s comment, we revised the manuscript.

If the amount of heat input for the same scan speed is highly determined (0.25~0.35 kJ/m), the cooling time will be relatively longer, which will affect the grain size. That is, as the amount of heat input increased, the grains were coarsened and that showed a decrease in hardness values. On the other hand, if the heat input supplied by the line energy is not sufficient (as the condition of 0.2 kJ/m), the internal unmelted zone may be widely distributed due to the insufficient melting and solidification of the fabricated powder, and the defects just below the surface may be present (as Fig. 2(a)), which may contribute to the large decrease in hardness. This can be found in the description of the properties of the microstructure in Figure 2(a).

Line 258 - You compare densities obtained under different conditions. Remove the 0.65 KJ/m because it use a different speed.

- According to the Reviewer’s comment, we revised the manuscript. Thanks for your comment.

Tab. 2 - Contains  a serious error, because there is not correlation between weight and density, although it should be.  The exceptions from tab. 2  must be correlated and explained otherwise  you violate the laws of  physics.

- We agree to the Reviewer’s opinion that there is no direct physical correlation between weight and density. Density measurement by Archimedes’ principle can be generally made by using air weighing and hydrostatic weighing (submerged in water). Table 2 showed the tested values by Archimedes’ density measuring method.

Lines 279 - 292 - The comments made do not have a statistical basis. In order to draw such conclusions, at least 9 specimens had to be  tried and processed in the same conditions, at each energy line. Repeat section 3. 3 under the conditions proposed above, based on  a reliable statistical analysis.

- According to the Reviewer’s comment, we revised the manuscript.

More than 15 specimen tests were performed per process condition under consideration of test errors and specimen soundness states. Data with a fatigue value of less than 400 MPa were not shown in the graph because they had no meaningful importance for high strength Ti64 alloy.

According to the reviewer's comment, the fatigue data of Fig. 6 were supplemented.

Line 297 - Explain what "...post heat treatment..." consisted of???

- According to the Reviewer’s comment, we revised the manuscript. When it is compared with the data of post heat treatment process (Ref. [28]), the HIP processing process is more effective than the post heat treatment process in order to significantly improve the high cycle fatigue properties of EBAM-printed products.

Fig. 6 - The data entered in the diagram are insufficient for statistical interpretation. This interpretation is mandatory if we take into account the strong random character of the formation, by local melting,  of the powders and their transformation into specimens. There is also a strong random character of the non-uniformities obtained between specimens that are not even analyzed. The above are supported even by the authors in the comments of the data presented in fig 7, without mentioning whether they are average of values or values obtained in a single measurement.

- According to the Reviewer’s comment, we revised the manuscript.

More than 15 specimen tests were performed per process condition under consideration of test errors and specimen soundness states. According to the reviewer's comment, the fatigue data of Fig. 6 were supplemented. Fig. 7 showed the fracture surfaces of the fatigue-fractured Ti6Al4V with different fabrication conditions, because the fracture surface cannot be observed in the unbroken specimen.

Conclusion - 1 -In the paper there not were : "...various scan speed...". It was a single 800mm/s. The ones of 500 mm/s is unconvincing because only one attempt was made with it. It would be normal that at a lower scanning speed (500 mm/s) and with a higher energy THAN OTHERS USED (0.65 kj/M) THESE SPECIMENS WOULD BE THE BEST MELTED, fact that does not result from he analysis of the presented data.

- There is wrong expression in the conclusion #1. According to the Reviewer’s comment, we revised the manuscript. Thanks for your comment.

General observations: The paper requires a better definition of variable parameters, a better explanation of internal phenomena and a better correlations between parameters and values obtained by using statistical analysis of experimental data, analysis imposed by the random nature of the processes that take place.

- According to the Reviewer’s comment, we revised it and improved the quality of paper. Thanks for your helpful comments.

Round 2

Reviewer 1 Report

Authors have correctly addressed all of my comments. 

Reviewer 2 Report

After revisions made the paper have now a good stile and content